# Extracting Heuristics from Large Language Models for Reward Shaping in Reinforcement Learning

## Abstract

Reinforcement Learning (RL) suffers from sample inefficiency in sparse reward domains, and the problem is further pronounced in case of stochastic transitions. To improve the sample efficiency, reward shaping is a well-studied approach to introduce intrinsic rewards that can help the RL agent converge to an optimal policy faster. However, designing a useful reward shaping function for all desirable states in the Markov Decision Process (MDP) is challenging, even for domain experts. Given that Large Language Models (LLMs) have demonstrated impressive performance across a magnitude of natural language tasks, we aim to answer the following question: *Can we obtain heuristics using LLMs for constructing a reward shaping function that can boost an RL agent's sample efficiency?* To this end, we aim to leverage off-the-shelf LLMs to generate a plan for an abstraction of the underlying MDP. We further use this LLM-generated plan as a heuristic to construct the reward shaping signal for the downstream RL agent. By characterizing the type of abstraction based on the MDP horizon length, we analyze the quality of heuristics when generated using an LLM, with and without a verifier in the loop. Our experiments across multiple domains with varying horizon length and number of sub-goals from the BabyAI environment suite, Household, Mario, and, Minecraft domain, show 1) the advantages and limitations of querying LLMs with and without a verifier to generate a reward shaping heuristic, and, 2) a significant improvement in the sample efficiency of PPO, A2C, and Q-learning when guided by the LLM-generated heuristics.

## 1 Introduction

Sample inefficiency of training Reinforcement Learning (RL) agents in sparse reward domains[1] has been a long-standing challenge (Ng et al., 1999; Laud & DeJong, 2003; Marthi, 2007; Grzes & Kudenko, 2008; Devlin & Kudenko, 2011). The number of environment interactions take a much severe hit if the domain further consists of stochastic transitions (Grześ, 2017; Ben-Porat et al., 2024). In an effort to improve this sample efficiency, reward shaping has been proven to be effective, which provides intrinsic rewards as a better training signal over just the sparse extrinsic (environment) rewards (Laud & DeJong, 2003; Marthi, 2007; Devlin & Kudenko, 2011).

Underlying reward shaping techniques (Ng et al., 1999; Devlin et al., 2011; Devlin & Kudenko, 2012; Gao & Toni, 2015; Eck et al., 2016), there is an inherent assumption made on how this reward shaping function can be constructed. One of the straightforward ways is for a domain expert to hand-engineer the reward shaping function, which can be cognitively demanding and additionally lead to a cognitive bias in the engineered rewards (Wu et al., 2024; Lightman et al., 2023). Yet another popular approach relies on learning the intrinsic rewards via Inverse RL (IRL) (Russell, 1998; Abbeel & Ng, 2004; Russell & Norvig, 2016), where the human records an expert demonstration for solving the task.

Lately, Large Language Models (LLMs) have shown remarkable performance spanning a wide variety of natural language-based tasks (Kocoń et al., 2023; Gilardi et al., 2023; Zhu et al., 2023) which can be attributed to the enormous and diverse data that they have been trained on. While some tasks have

---

[1]We consider the case where the agent gets +1 reward at the goal state, and 0 otherwise.

benefited from prompting off-the-shelf LLMs (Bubeck et al., 2023; Bhattacharjee et al., 2024), others that are either highly domain-specific or require higher degrees of generalizability, require fine-tuning (Li et al., 2024; Yang et al., 2024). However, several recent studies have shown the performance of prompting LLMs directly to be brittle and unreliable (Valmeekam et al., 2023; Stechly et al., 2024; Verma et al., 2024). Similarly, the latter approach is bottlenecked by the need for sufficient task-specific data and expensive computation required for LLM fine-tuning. Yet, they continue to show some promise when tasked with solving a sufficiently relaxed version of the original problem (Nirmal et al., 2024), or assisting in obtaining the final solution (Kambhampati et al., 2024). Keeping this trade-off in mind for our use case, we aim to answer: *Can we obtain heuristics using LLMs for constructing a reward shaping function that can boost an RL agent's sample efficiency?*

From the limited exploratory works that currently lie at utilizing LLMs for guiding RL (Liang et al., 2023; Du et al., 2023; Carta et al., 2023; Jiang et al., 2019; Kwon et al., 2023; Wang et al., 2023; Ma et al., 2023), LLMs have particularly been effective in providing either high-level (hierarchical) policy guidance (Jiang et al., 2019; Liang et al., 2023) or the reward function (Kwon et al., 2023; Ma et al., 2023), which may only be feasible for tasks where there has been sufficient background knowledge or data that could have possibly been part of the language model's training data. However, treating LLMs as sources of approximate common-sensical knowledge, our intuition is that we can expect them to generate a heuristic that can be useful for the downstream task (Cheng et al., 2021). To this end, we take inspiration from the problem abstraction methods, which have been extensively studied in the planning and RL literature (Dieterich et al., 1998; Sutton et al., 1999; Lane & Kaelbling, 2002; Kattenbelt et al., 2010; Kulkarni et al., 2016; Gopalan et al., 2017; Jiang et al., 2019; Nashed et al., 2021), and query LLMs to obtain a solution for a sufficiently relaxed abstract problem.

For a given RL problem, the desired heuristic can be obtained at both, the low-level, and a high-level (symbolic) action space. For example, for the *Household* environment shown in Figure 5, low-level actions comprise ⟨ up, down, left, right ⟩, whereas high-level (symbolic) actions comprise ⟨ pickup key, open door ⟩, etc. Furthermore, the choice of abstraction depends on the nature of the underlying problem, since LLMs operate in the text space. One straightforward possibility is to consider a deterministic abstraction of the underlying stochastic Markov Decision Process (MDP) (Yoon et al., 2008). While the abstract MDP in this case will still require a solution using the low-level action space, it may only be useful for the RL agent if LLMs can easily find a goal-reaching plan for that deterministic problem. Hence, we first investigate the usefulness of directly prompting off-the-shelf LLMs for the deterministic abstraction of short horizon stochastic MDP problems, which tend to yield incomplete plans that are consistently unable to reach the goal. Furthermore, our experiments show the ineffectiveness of using these incomplete plans as heuristics for reward shaping the RL agents. While a deterministic abstraction on the low-level action space still remains a challenging planning problem for LLMs, a yet another possibility particularly for long-horizon problems, is to construct a hierarchical abstraction of the underlying MDP which consists of a high-level (symbolic) action space (Jiang et al., 2019; Liang et al., 2023). A plan in this hierarchical abstraction will include some or all the sub-goals the agent has to achieve in the correct order to reach the goal, that can further be used to design our reward shaping function accordingly. Note, that our work does not intend to pose a dichotomy between a deterministic and a hierarchical MDP setup, but aims to study these two possible MDP abstractions for a downstream RL task which has sparse-reward and stochastic transitions.

While LLMs may still not be able to yield valid (executable) plans each time for the abstract problem, we further investigate the performance when LLMs are used with a verifier in the loop. The presence of such a verifier can help generate a valid goal-reaching plan. The idea of such verifier-augmented LLM setups has also been recently seen for planning and reasoning problems(Kambhampati et al., 2024; Gundawar et al., 2024; Liang et al., 2023; Ma et al., 2023; Wang et al., 2023). Finally, we leverage the LLM and LLM+verifier generated plans as a basis for constructing the reward shaping function for the downstream RL sparse reward task. Through this work, we aim to understand the possible role of LLMs as heuristic generators for downstream RL problems, and discuss important trade-offs that need to be considered for choosing the right abstraction and constructing a verifier.

The contributions of this work can be summarized as follows:

1. In the context of investigating LLMs' utility in generating heuristics, we study two different types of MDP abstractions - *deterministic* and *hierarchical*, for short and long-horizon problems respectively.

2. For both types of abstractions, we investigate the performance of LLMs in generating heuristics with and without the presence of a verifier. Utilizing the LLM and LLM+verifier generated heuristics, we construct a reward shaping function for the underlying sparse-reward MDP.

3. With experiments on the BabyAI environment suite, Household, Mario, and, Minecraft domains, we show a significant boost in the sample efficiency of RL algorithms by demonstrating results with PPO, A2C, and, Q-learning algorithms.

For the rest of the paper, we begin with situating our work in the domain of LLM-guided RL works and give a brief background of the respective literature in Section 2. Next, we provide the preliminaries and formally define our problem statement in Sections 3, and discuss our investigations of using LLMs as heuristics for reward shaping in detail in Section 4, followed by experiments and results in Section 5. We also include a discussion on the key takeaways from this work on the choice of abstraction and the utility of having verifier-augmented LLMs for obtaining heuristics. Finally, we conclude the work in Section 6. An appendix with additional experiment details has also been attached, and code will be released on acceptance.

## 2 RELATED WORK

**Sparse Reward RL and LLM-based Guidance:** The seminal foundational work by (Ng et al., 1999) provided policy invariance guarantees using Potential-based Reward Shaping (PBRS) for boosting the sample efficiency of RL agents, followed by further theoretical investigations by (Laud & DeJong, 2003; Wiewiora, 2003). (Pathak et al., 2017) also showed the advantages of using intrinsic rewards for training RL agents. Reward shaping methods have been studied under several dimensions, including but not limited to automatic reward learning (Grzes & Kudenko, 2008; Marthi, 2007), (Zhang et al., 2024; Srivastava et al., 2024), multi-agent domains (Devlin & Kudenko, 2011; Sun et al., 2018), meta-learning (Zou et al., 2019), etc. Primarily, the sources of obtaining and/or learning intrinsic rewards includes domain experts hand-engineering the rewards (Wu et al., 2024; Lightman et al., 2023), via providing feedback (Lee et al., 2023), or via providing expert demonstrations (Argall et al., 2009). More recently, Large Language Models have been utilized to give feedback on RL agent's environment interaction (Du et al., 2023; Ma et al., 2023; Kwon et al., 2023; Cao et al., 2024) or directly provide the reward function for the RL agent's task.

**LLMs for Planning and Search:** There is a research divide in the current literature regarding the planning, reasoning and verification abilities of Large Language Models. While popular works claiming LLM reasoning abilities have proposed several prompting methods (Wei et al., 2022; Yao et al., 2023; Long, 2023; Yao et al., 2024; Besta et al., 2024), there have been independent investigations refuting such claims using LLMs for solving deterministic planning and classical reasoning problems (Valmeekam et al., 2023; Stechly et al., 2024; Verma et al., 2024). While LLMs are themselves not reliable for providing accurate feedback (Stechly et al., 2024), other than in natural language tasks (Yao et al., 2023), recent works have augmented LLMs with task-specific verifiers (for example, a Python compiler that can check LLM-generated code for syntactic correctness) that can evaluate the validity (not necessarily correctness) of the LLM-generated output and provide feedback to the LLM accordingly (Kambhampati et al., 2024; Ma et al., 2023; Wang et al., 2023; Liang et al., 2023). These augmented LLM frameworks have been shown to be useful for improving the overall task performance when prompting LLMs. We further utilize these plans to construct the reward shaping function for the downstream RL agent.

## 3 PRELIMINARIES AND PROBLEM STATEMENT

We consider a finite horizon Markov Decision Process (MDP) $\mathcal{M}$ defined by the tuple ($\mathcal{S}$, $\mathcal{A}$, $\mathcal{P}$, $\mathcal{R}$, $\gamma$). Here, $\mathcal{S}$ represents the set of all possible states, $\mathcal{A}$ represents the set of all possible actions, $\mathcal{P} : \mathcal{S} \times \mathcal{A} \rightarrow \mathcal{S}$ is the stochastic state transition function where $\mathcal{P}(s'|s, a)$ is the transition probability for $s, s' \in \mathcal{S}$ and $a \in \mathcal{A}$, $\mathcal{R} : \mathcal{S} \times \mathcal{A} \times \mathcal{S} \rightarrow \mathbb{R}$ is the reward function, and $\gamma$ is the discount factor. In our case, we consider an MDP $\mathcal{M}$ with sparse rewards, i.e., $\mathcal{R} = 1$ for $g \in \mathcal{S}$ and $\mathcal{R} = 0$ otherwise, where $g$ is the goal or termination state. The objective of the agent is to learn a

parameterized policy $\pi_\theta(a|s)$ which maximizes the discounted cumulative reward for the trajectory $\tau$, $\mathcal{J}(\theta) = \mathbb{E}_{\tau \sim \pi_\theta} \left[ \sum_{t=0}^{T} \gamma^t r_t \right]$. In the paper, we consider two types of MDP abstractions:

**Deterministic Abstraction:** For short-horizon problem setting, we consider the MDP $\mathcal{M}'$ modified from the underlying stochastic sparse-reward MDP $\mathcal{M}$ such that, $\mathcal{M}'$ can be defined using the tuple $(\mathcal{S}, \mathcal{A}, \mathcal{T}', \mathcal{R}, \gamma)$, where $\mathcal{T}' : \mathcal{S} \times \mathcal{A} \rightarrow \mathcal{S}$ is the deterministic transition function. We aim to obtain a guide plan $\pi_g$ in $\mathcal{M}'$ using the LLM-environment interaction, that can further be used as a heuristic to construct a reward shaping function for the underlying RL problem for learning $\pi_\theta$.

**Hierarchical Abstraction:** For long-horizon problem setting, we consider the abstract MDP $\mathcal{M}''$ modified from the underlying stochastic sparse-reward MDP $\mathcal{M}$ such that, $\mathcal{M}''$ can be defined in the form of declarative action-centered representation of a planning task. Specifically, we consider a STRIPS-style planning problem (Fikes & Nilsson, 1971), where the planning model can be defined in the form $\mathcal{P} = \langle F, A, I, G \rangle$. $F$ is a set of propositional state variables or fluents defining the hierarchical state space. Similar to (Guan et al., 2022), we assume access to a function $\mathcal{H} : \mathcal{S} \times F \rightarrow \{0, 1\}$ that maps MDP states to hierarchical fluents, such that $\mathcal{H}(s, f)$ is set to true for fluent $f$ if $f$ exists in the MDP state $s \in \mathcal{S}$. $A$ is the set of action definitions, where action $a \in A$ is defined as $a = \langle prec^a, add^a, del^a \rangle$; where $prec^a$ are set of preconditions in the form of binary features that need to be true in a state to execute action $a$, and, $add^a$ and $del^a$ are the add and delete effects that capture the set of binary features set to true and false, respectively, when action $a$ is executed. For this work, since $A$ represents the high-level (symbolic) actions for us, we will refer to these actions as $\mathcal{A}^h$ to distinguish them from the underlying MDP's low-level action space represented by $\mathcal{A}$. Finally, $I$ is the initial state of the agent and $G \subseteq F$ is the goal specification. A solution to the planning problem $\mathcal{P}$ is the set of correctly-ordered actions which will act as the guide plan $\pi_g$. Similar to the deterministic abstraction case, $\pi_g$ can further be used to construct as a heuristic to construct a reward shaping function for the underlying RL problem for learning $\pi_\theta$.

**Problem Statement:** In this work, we first aim to obtain the guide plan $\pi_g$ by querying a LLM which can act as the heuristic to construct a reward shaping function for the underlying stochastic sparse-reward MDP $\mathcal{M}$. Once we have obtained $\pi_g$, we adopt the Potential-based Reward Shaping approach, as proposed in (Ng et al., 1999). In the case of deterministic abstraction, the goal is to obtain $\pi_g$ using a LLM where $\pi_g$ is a sequence of (state, action) pairs where the state and actions are same as the underlying MDP $\mathcal{M}$. Since $\pi_g$ consists of (state, action) pairs which are same as the underlying MDP $\mathcal{M}$, we construct a (state, action)-based reward shaping function $\mathcal{F} = \Phi(s', a') - \Phi(s, a)$ for $s, s' \in \mathcal{S}$ and $a, a' \in \mathcal{A}$ (Wiewiora et al., 2003) and $\Phi$ is the potential function which gives a potential to any $(s, a)$ if a (state, action) pair exists in $\pi_g$. In the latter case of hierarchical abstraction, the goal is to obtain $\pi_g$ using a LLM where $\pi_g$ is a sequence of PDDL actions, which represent the different sub-goals that exist in the underlying MDP $\mathcal{M}$. Hence, similar to (Grzes & Kudenko, 2008), we adopt the state-based potential function $\mathcal{F} = \Phi(s') - \Phi(s)$ for $s, s' \in \mathcal{S}$ where the state potential is given by the number of sub-goal (or landmark) fluents that have been satisfied in $\pi_g$.

# 4 INVESTIGATING LLM-GENERATED HEURISTICS FOR REWARD SHAPING

In this section, we first discuss the prompt setup for directly querying off-the-shelf LMs to obtain the guide plan $\pi_g$ (Section 4.1), and then discuss the verifier construction and prompt setup for the verifier-augmented LLM framework (Section 4.2). Lastly, we discuss how we construct the reward shaping function using $\pi_g$ for both types of abstractions in Section 4.3.

## 4.1 DIRECTLY PROMPTING LLMS WITHOUT VERIFICATION

### 4.1.1 PROMPT CONSTRUCTION

We consider the zero-shot setting for prompting the LLM to obtain $\pi_g$. For the deterministic MDP $\mathcal{M}'$, consider the example of the BabyAI *DoorKey* environment in Figure 1, where the task of the agent is to "use the key to open the door and then get to the goal". We construct the LLM prompt with three components. The first is the *Task Description* that defines the task that the LLM (as the agent in this case) has to achieve. In this example, we specify that the environment is a 3x3 maze

Figure 1: (I) We use the verifier-augmented LLM to generate a valid (guide) plan for the relaxed search problem. (II) We construct the reward shaping function using the guide plan to add intrinsic rewards by updating the RL agent's replay buffer. (III) Using these intrinsic rewards, the RL agent learns an optimal policy for the underlying stochastic sparse-reward MDP.

that consists of objects such as a key, a door, walls, and a goal location. We further include the goal of the agent that is obtained from the environment, followed by the environment's action space ($\mathcal{A}$). Next, we include *Observation Description* which describes the current view of the environment. We admit that representing spatial relationships in text as an input to an LLM can be a challenging task, a problem that is exacerbated for larger and more complex environments. Similar to recent works that have attempted to prompt LLMs with spatial descriptions (Patel & Pavlick, 2021), we represent the 3x3 grid as three rows of objects that are currently observed in the environment state ($s_i$). Finally, the third and the last component of this prompt is the *Query Description* which poses the question to the LLM to guess the set of actions that the agent should take in the environment (Appendix D.1).

For the hierarchical MDP $\mathcal{M}''$, we are able to directly tease out the PDDL model of the underlying task using LLMs as shown by (Guan et al., 2023). Now, since we have access to the PDDL domain model and the problem specification of the abstract MDP, we can relax the requirement to convert the environment observation in to spatial text representations. The *Observation Description* only consists of the PDDL domain and problem specification, and thus, we do not need to provide the *Task Description* separately. Finally, the *Query Description* is included which poses the question to the LLM to guess the set of high-level (symbolic) actions for the given problem. Note, that while the PDDL model teased from the LLMs can be noisy, we are able to ease out on the prompt engineering efforts with the presence of such a representation, which were otherwise required for the deterministic abstraction case as mentioned above (Appendix D.1).

## 4.2 AUGMENTING LLMs WITH A VERIFIER

In this subsection, we discuss the verifier-augmented LLM framework which is used to generate a valid solution, i.e., $\pi_g$, for the abstract MDP. This plan is further used to construct a reward shaping function, resulting in a sample efficiency boost for the RL agent that learns a policy on the original stochastic, sparse reward problem.

### 4.2.1 VERIFIER CONSTRUCTION

For the deterministic MDP $\mathcal{M}'$, we will again use the BabyAI *DoorKey* environment, as shown in Figure 1, as the example to discuss the details of how we construct the verifier. It is important to note that we eventually require a set of actions from this verifier, that are feasible at any given state of the environment that the agent can be in. We will refer to this set of valid actions as $\{\mathcal{A}_v|s\}$ and set of invalid actions as $\{\mathcal{A}_{v-}|s\}$ for $s \in \mathcal{S}$. Hence, for the BabyAI *DoorKey* environment, we begin with computing the agent's position ($x$, $y$) at the given state ($s_i$). At any given step of the LLM-environment interaction, we can verify each action that the agent can and can not take for a given state, i.e., either $a \in \{\mathcal{A}_v|s_i\}$ or $a \in \{\mathcal{A}_{v-}|s_i\}$. For example, in the state ($s_i$) shown in Figure 1, the agent has already picked up the key but can only move forward if the door is open. Hence, the model-based verifier, given the agent's current position ($x$, $y$) and the set of valid actions $\{\mathcal{A}_v\}$ taken until step $i$, computes the set of actions that are feasible in the current state, i.e., turn left, turn right, and toggle (open door). Next, given the LLM's guessed action, i.e., move forward,

the verifier finds the action to be infeasible and generates a back-prompt that is appended to the LLM's original prompt. We further discuss prompt construction details in Section 4.2.2.

For the hierarchical abstraction case, consider the abstract MDP $\mathcal{M}''$ for the same BabyAI *DoorKey* environment shown in Figure 1. In this case, our PDDL actions can be ⟨ pickup_key, open_door, reach_goal ⟩, representing the sub-goals that the underlying MDP agent needs to achieve. If the agent is in the initial state, the actions ⟨ open_door, reach_goal ⟩ will be part of the invalid action set $\{\mathcal{A}_{v-}|s\}$, and the model-based verifier will generate a back-prompt. Hence, the verifier in this case is much more simplified as it is only required to check if the LLM generates syntactically correct actions, and if the actions (sub-goals in this case) are output in the correct order as per the environment constraints.

### 4.2.2 PROMPT CONSTRUCTION

At any given step $i$ in the LLM's interaction with the environment for the deterministic abstraction case, we once again construct the LLM prompt with three components. The first is the *Task Description* followed by the *Observation Description* that is now generated independently at every step after the previous LLM-generated valid action is executed, and finally include the *Query Description* asking the LLM to guess the next low-level action. We refer to this as the *step-prompt* for our discussion. Once the LLM returns a valid action as a guess to the *step-prompt*, i.e. $\pi_{LLM}(s_i) = a$ for $a \in \{\mathcal{A}_v|s_i\}$, we execute the action in the environment and continue the interaction. Consider the case where the LLM has guessed an invalid action, i.e. $\pi_{LLM}(s_i) = a$ for $a \in \{\mathcal{A}_{v-}|s_i\}$. In this case, we do not execute the action in the environment, but rather, construct the *back-prompt* by appending the feedback given by the model-based verifier to the current prompt. To the existing *step-prompt*, we add this verifier feedback that lists all the invalid actions ($\{\mathcal{A}_{v-}|s_i\}$) guessed by the LLM for the current state $s_i$ and prompt the LLM again to choose a different action that is not in $\{\mathcal{A}_{v-}|s_i\}$. Once the LLM guesses a valid action, we break out of this back-prompting loop and continue our environment interaction as mentioned above (see Appendix D.2 for the complete prompt).

For the hierarchical abstraction case, we prompt the LLM with the PDDL domain and problem specification in the *Observation Description*, and query it to generate only the next high-level action ($\in \mathcal{A}^h$) in the *Query Description*. Additionally, we provide the LLM-generated plan so far, i.e., $\pi_g^{i-1}$ consisting of the valid actions till step $i-1$. Similar to the deterministic abstraction case, we *back-prompt* the LLM if it outputs an invalid action, and continue the interaction till it finds a valid plan (see Appendix D.2 for the complete prompt).

### 4.3 REWARD SHAPING USING $\pi_g$

Potential-based Reward Shaping (PBRS) (Ng et al., 1999) allows for injecting intrinsic rewards for training an RL agent on a sparse reward problem while guaranteeing the policy invariance property. Once we have obtained $\pi_g$ by querying the LLM with and without a verifier, we utilize it to construct our reward shaping function $\mathcal{F}$. Using the reward shaping function $\mathcal{F}$, we assign potentials to the $(s, a)$ pairs in the LLM-generated plan for the deterministic case, and to the states $(s)$ in the LLM-generated plan for the hierarchical case. Formally, we adopt the definition which utilizes a shaping reward function $\mathcal{F}$ such that our updated reward function can be defined as: $\mathcal{R}'(s, a) = \mathcal{R}(s, a) + \mathcal{F}(s, a)$ for $s \in \mathcal{S}, a \in \mathcal{A}$ in the former case, and as $\mathcal{R}'(s) = \mathcal{R}(s) + \mathcal{F}(s)$ in the latter. We refer the readers to Section 5 for details on $\mathcal{F}$ for both types of abstractions.

In the exploration phase, the RL agent stores the environment interactions, i.e., $(s, a, s', r)$ tuples, in a dataset buffer $\mathcal{D}$. Given $\mathcal{F}$, we update the buffer $\mathcal{D}$ with the shaped rewards such that $(s, a, s', r) \rightarrow (s, a, s', r')$ where $r'$ is the shaped reward. Our RL agent then continues the learning over this updated dataset buffer $\mathcal{D}'$. We present the complete pipeline in Figure 1 and Algorithm 1 in Appendix C.

## 5 EXPERIMENTS & RESULTS

We first aim to investigate the quality of the heuristics generated by the LLMs in the form of $\pi_g$, with and without a verifier in the loop. Next, we construct the reward shaping function using $\pi_g$ for both, the deterministic and the hierarchical abstraction cases. Finally, we utilize the reward shaping function to inject intrinsic rewards into the RL agent's training loop and measure the boost in sample

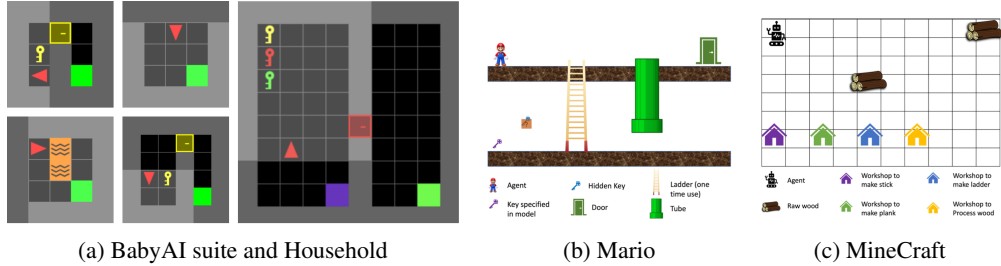

(a) BabyAI suite and Household      (b) Mario      (c) MineCraft

Figure 2: Visualizations for the BabyAI suite and Household, Mario, and, the MineCraft environment.

efficiency. Hence, we aim to answer the following: **RQ1:** How do LLM and LLM+verifier-generated plans compare in terms of the quality of obtained heuristics? In RQ1 results, we observe LLMs generating, both partial and complete (goal-reaching) plans. Hence, for comparing the effectiveness of these obtained heuristics for the underlying RL problem, we aim to answer the following: **RQ2:** How effective is the reward shaping with the use of partial and complete plans for boosting the RL sample efficiency?

### 5.1 EVALUATION DOMAINS

For our empirical evaluations on the short-horizon setting with stochastic transitions, we utilize the BabyAI suite of environments as shown in Figure 2a, namely - *DoorKey* that requires the agent to pick up the key to open a door and reach the goal; *Empty-Random* where there are no obstacles but the initial position of the agent is randomized for each episode; and finally, *LavaGap* in which the agent has to reach the goal location while avoiding the adversarial objects (the lava tile) present in the environment. For each of these environments, we construct the deterministic abstraction to query the LLM for obtaining the guide plan $\pi_g$ consisting of the low-level actions from the underlying MDP.

For the long-horizon setting with stochastic transitions where we consider the hierarchical abstraction, we utilize - the *Household* environment which is a more complex version of the *DoorKey* environment (Figure 2a) and requires the agent to pick the right key for unlocking the door and reaching the goal; the *Mario* environment (Figure 2b) where the agent needs to go down the green tube, pick both the keys, climb up the ladder and reach the door; and finally, the *MineCraft* environment ((Figure 2c) where the agent needs to first collect both the pieces of raw wood, go to the workshop to process wood, make stick and plank at the respective workshops using the two processed woods, and go to the workshop to make ladder using the stick and the plank.

### 5.2 COMPARING THE LLM-GENERATED HEURISTICS (RQ1)

In order to study RQ1 for both abstractions, we run all our experiments using four different LLM models. We highlight the experimental setup, baselines and evaluation metrics in this sub-section.

**Experimental Setup:** For the deterministic abstraction (RQ1.1), we run all our LLM-based experiments on the *DoorKey*, *Empty-Random*, and, the *LavaGap* environments. In the case of directly prompting LLMs, we set the temperature to 0.5, and instruct the LLM to generate the entire plan at once. For the hierarchical abstraction case (RQ1.2), we use the same settings for running our experiments on the *Household*, *Mario*, and, the *MineCraft* domain.

**Evaluation Metrics:** We compare the (mean $\pm$ standard deviation) for the generated plans in terms of the length of a successful plan and total rewards for RQ1.1; and the number of sub-goals achieved for RQ1.2.

**Results:** From the results in Table 1, we note that direct LLM prompting can not generate a valid plan in all but two cases across the three environment settings in the deterministic abstraction (low-level action space). For our LLM+verifier framework, we obtain a goal-reaching plan for both *Empty-Random* and *LavaGap* environments across all LLMs, but only `gpt-3.5-turbo` is able to successfully solve the task for the *DoorKey* environment. For the verifier-augmented setup, we

Table 1: RQ1.1 results: plan length and rewards are averaged across 3 runs. Numbers in parenthesis refer to scores from successful runs via directly prompting LLMs for the specific LLM-Environment pair. For all other cases, the direct prompting LLM runs failed to produce valid plans, as also seen for certain LLM+verifier cases.

| Environment | Metric | GPT-3.5 | GPT-4o | Claude Haiku | Llama 3 8B |
|---|---|---|---|---|---|
| DoorKey | Avg. plan length | $15.2 \pm 6.57$ | – | – | – |
| | Avg. reward | $0.9452 \pm 0.02$ | – | – | – |
| Empty-Random | Avg. plan length | $20 \pm 6.08$ $(5 \pm 0)$ | $4.33 \pm 0.577$ $(5 \pm 0)$ | $13 \pm 0$ | $15.67 \pm 6.429$ |
| | Avg. reward | $0.82 \pm 0.054$ $(0.955 \pm 0)$ | $0.961 \pm 0.005$ $(0.955 \pm 0)$ | $0.883 \pm 0$ | $0.859 \pm 0.057$ |
| LavaGap | Avg. plan length | $16 \pm 8$ | $13.33 \pm 7.57$ | – | $20 \pm 8.48$ |
| | Avg. reward | $0.8559 \pm 0.07$ | $0.879 \pm 0.068$ | – | $0.82 \pm 0.076$ |

Table 2: RQ 1.2 results: fraction of sub-goals reached is averaged across 3 runs.

| Environment | Variant | GPT-3.5 | GPT-4o | Claude Haiku | Llama 3 8B |
|---|---|---|---|---|---|
| Household | *vanilla* | 0.2 | - | 1 | 0.732 |
| | *with verifier* | 0.132 | 0.266 | 0.466 | 0.2 |
| Mario | *vanilla* | 0.25 | - | 1 | 1 |
| | *with verifier* | - | 0.25 | 1 | 0.665 |
| Minecraft | *vanilla* | 0.25 | - | 1 | 0.665 |
| | *with verifier* | - | 1 | 0.75 | 0.5 |

also note that the LLM repeats a set of valid but incorrect actions for multiple steps and exhausts the query budget for the number of step-prompts before reaching the goal. From the results in Table 2, we note that direct LLM prompting is also able to generate complete plans, and that a step-by-step verifier in the loop may not always guarantee improvements. [2]

## 5.3 Evaluating the Sample Efficiency boost in RL training (RQ2)

To study RQ2, we select the RL algorithms that have been used in the literature for the respective environments. In the deterministic abstraction case (RQ2.1), we train PPO and A2C algorithms on each environment layout of the BabyAI suite (Chevalier-Boisvert et al., 2018); and train Q-learning for the *Household*, *Mario*, and the *MineCraft* environment (Guan et al., 2022) in the hierarchical abstraction case (RQ2.2). We provide all training and hyperparameters details in Appendix C.

**Experimental Setup:** Recall, from Section 5.2, that we generate the guide plan $\pi_g$ for each environment layout. In RQ2.1, we train the PPO and A2C agents on our underlying stochastic sparse-reward problem, with the reward shaping function $\mathcal{F}$ constructed using each of these guide plans. Following the look-back advice principle for reward shaping in PPO and A2C algorithms (Wiewiora et al., 2003), the reward shaping function can is defined as $\mathcal{F}(s_t, a_t, s_{t-1}, at - 1) = \Phi(s_t, a_t) - \gamma^{-1}\Phi(s_{t-1}, a_{t-1})$. In RQ2.2, we train Q-learning with the state-based reward shaping function $\mathcal{F}$ which is defined using the number of fluents that are true in any given state as shown by (Grzes & Kudenko, 2008).

**Baselines & Evaluation Metrics:** For RQ2.1, we adopt the RL baselines for both PPO and A2C algorithms from (Chevalier-Boisvert et al., 2018). For RQ2.2, we adopt the Q-learning baseline for each of the three environments from (Guan et al., 2022). We further include a reward shaping baseline achieving only a single sub-goal in each environment, that we obtain using the LLM-generated (partial) plan without a verifier in the loop. While we observe that directly prompting LLMs in the

---

[2] We suspect that one possible reason for this behavior could be that for some LLMs, it may be easier to get better performance if queried for the entire solution plan at once than step-by-step.

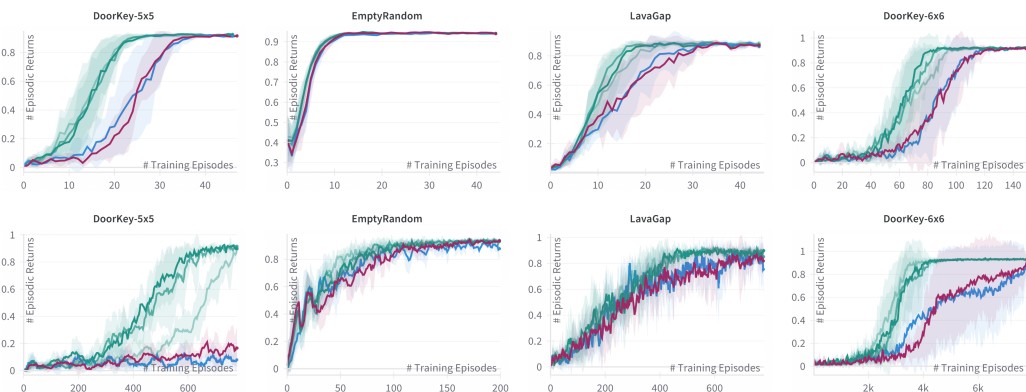

Figure 3: **RQ2.1 Results:** Smoothed learning curves comparing vanilla PPO (*top*) and vanilla A2C (*bottom*) with reward shaping on respective algorithms using LLM-generated partial plan and with reward shaping using three variations of LLM-generated complete plans, as measured on the episodic returns. The solid lines and shaded regions represent the mean and standard deviation across five runs, respectively.

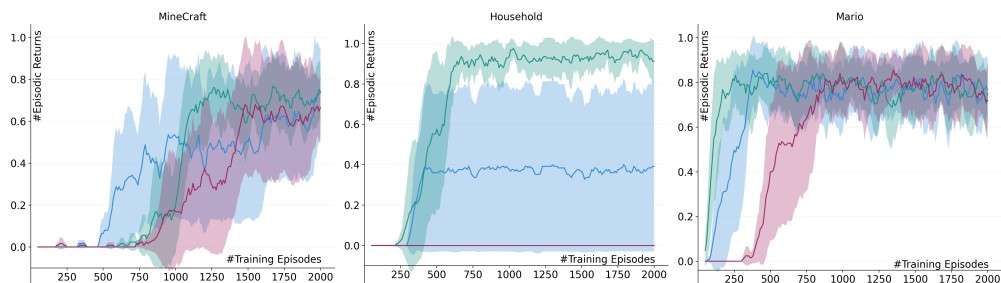

Figure 4: **RQ2.2 Results:** Smoothed learning curves comparing baseline Q-learning training against baseline Q-learning with reward shaping using LLM-generated partial plan, and with LLM-generated complete plan as measured on the episodic returns. The solid lines and shaded regions represent the mean and standard deviation across five runs, respectively.

hierarchical abstraction case (Table 2) is also able to generate complete plans, we want to specifically test if a LLM-generated partial plan in hierarchical abstraction can be useful for reward shaping compared to the case of a LLM-generated partial plan in the deterministic abstraction. For each experiment, we plot episodic returns against number of training episodes in Figure 3 and Figure 4.

**Results:** From results shown in Figure 3 for RQ2.1, we note the most significant boost in sample efficiency due to the reward shaping using our augmented LLM-generated plan in the BabyAI *DoorKey-5x5* environment, followed by *LavaGap*, and results in *Empty-Random-5x5* environment are similar to the vanilla RL baseline. For the *LavaGap* environment, the difference between reward-shaped policy training and the baselines is relatively smaller than that in *DoorKey-5x5*. One possible reason here could be that while the action space for this environment is smaller (three actions only), using the LLM-generated plan for reward shaping allows the RL agent to learn to avoid the lava tiles faster and reach the goal location. For the results shown in Figure 4 for RQ2.2, we use reward shaping using a LLM-generated (partial) plan which satisfies only one sub-goal for each environment.[3] The training curves for reward-shaped Q-learning using LLM-generated complete plans significantly outperform the other settings. However, unlike RQ2.1 results, reward shaping using partial plans also outperforms the Q-learning baseline in terms of episodic returns and earliest policy convergence. We conclude that, partial plans in the high-level (symbolic) action space can be more useful heuristics as compared to partial plans in the low-level action space. We further analyze these results in the next sub-section.

---

[3]There are 5 sub-goals in *Household*, and 4 in the *Mario* and *MineCraft* domains.

## 5.4 DISCUSSION

In this sub-section, we aim to draw insights from our experiments for both RQ1 and RQ2. Specifically, we discuss the trade-offs a) between the utility and the ease of constructing a deterministic or a hierarchical abstraction, and b) between the quality of heuristic obtained and effort to construct a verifier for querying LLMs.

**Deterministic or Hierarchical Abstraction?** Recall, that the primary difference between the two types of abstractions lies in the state and action space representations. The deterministic abstract MDP $\mathcal{M}'$ preserves the same state and low-level action space representation as the underlying stochastic sparse-reward MDP, while the hierarchical abstraction only considers the high-level sub-goals over the underlying MDP task. The utility of each of the two types stems from how well LLMs perform to generate the abstract plan. Our analysis from RQ1.1 shows that LLMs are unable to consistently generate a solution when we represent the underlying MDP problem as part of a text prompt. Also, there is an added burden due to the prompt engineering effort required in the deterministic abstraction case due to the preserved state and action spaces. However, as we can provide shaped rewards to each (state, action) pair in the goal-reaching plan generated for the deterministic MDP, the reward shaping signal for the downstream RL agent is much more fine-grained. In the hierarchical abstraction case, even when LLMs are able to find a partial solution in terms of correctly ordered sub-goals, we are able to provide intrinsic rewards to states where those sub-goal fluents are true making LLM-generated heuristics useful even in the absence of a verifier. As pointed out by (Kambhampati et al., 2024), while we could use PDDL planners to get this abstract plan, these planners are useful in a narrow set of domains whereas LLMs can be useful in many more generalizable scenarios. Moreover, we believe that extending our reward shaping work to partially-observable state space and continuous action space environments can be a valuable future extension. For example, our hierarchical abstraction can be easily utilized for continuous action spaces. Similar to the discrete action space experiments shown for Mario, Minecraft, and the Household environment in our work, the RL agent can learn a policy that follows the correct sequence of subgoals using shaped rewards in a continuous environment too. To conclude, we note that while the reward shaping signal may not be as fine-grained, hierarchical abstraction allows for obtaining better reward shaping heuristics using LLMs.

**How useful is having a verifier in the loop?** For either type of the abstraction, the presence of a verifier allows LLMs to use a generate-test-verify loop for generating an improved plan. However, note that the verifier assumes the knowledge of the environment's domain model such that it can check for any valid or invalid actions at any given step of the LLM-environment interaction. This requirement is slightly relaxed as the LLMs can be useful translators in obtaining a noisy approximate domain model when provided with natural language instructions (Guan et al., 2023). Hence, in the case of hierarchical abstraction, the verifier is much simpler than that in deterministic abstraction as it only needs to check if the preconditions of the LLM-generated sub-goal are satisfied. To conclude, we note that if obtaining the domain model is not expensive, having a verifier in the loop can improve the quality of heuristic obtained. Future works can delve into alternative methods of constructing these verifiers.

## 6 CONCLUSION & FUTURE WORK

Designing reward shaping methods to inject intrinsic rewards useful for training Reinforcement Learning agents can be challenging, even for domain experts. Lately, Large Language Models have shown remarkable success in a variety of natural language tasks, while also encountering limitations when it comes to prompting them for planning and reasoning problems. Keeping these limitations in mind, we aim to investigate the utility of LLMs to generate heuristics for guiding RL training in sparse reward tasks. We study a deterministic and a hierarchical abstraction of the underlying MDP for short and long-horizon environments, respectively. Using LLM and LLM+verifier-generated plans, we construct a reward shaping function for the underlying MDP. Our results indicate a boost in the sample efficiency of downstream RL training across BabyAI suite, Household, Mario and Minecraft domains. Finally, we discuss the trade-offs associated with the choice of abstraction and the presence of a verifier for obtaining heuristics that can guide RL agents.

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
