# OpenReview forum: "Extracting Heuristics from Large Language Models for Reward Shaping in Reinforcement Learning"
_ICLR.cc/2025/Conference — ICLR 2025 Conference Withdrawn Submission_

### Official Review · Reviewer_JNWs · 2024-10-29

**Soundness:** 3
**Presentation:** 2
**Contribution:** 2
**Rating:** 3
**Confidence:** 3

**Summary:**

The authors introduce a method for reward shaping that leverages plans generated by Large Language Models across three sets of domains and at two levels of abstraction and with an added component of plan verification for one experimental condition. Given a textual description of the environment (which is derived using two separate methods for the deterministic and hierarchical abstraction settings), LLMs are asked to produce a guide-plan, a sequence of actions it guesses will solve the MDP. Using this guide-plan, potential-based reward shaping (PBRS) is used to generate a shaping reward function, which is used to train a variety of RL agents on three sets of environments (including A2C, PPO, and Q-Learning). In another experimental condition, a verifier is used to validate whether the planning actions generated by the LLM are valid, and the LLM is re-prompted if they are not.

**Strengths:**

- The paper introduces a novel synthesis of LLMs, planning, and reward shaping that, to my knowledge, has not been done before.
- A large number of environments are used to evaluate the method.
- More than one RL algorithm is used to assess the value of the shaped reward function.
- A variety of LLMs are used to assess their abilities to make plans.

**Weaknesses:**

While the paper presents a novel idea, the pipeline provided seems quite contrived and the presentation of the ideas is slightly confusing and not well-placed in the surrounding literature. Here are a few points of criticism:
1. LLMs are prompted for guide-plans, and verifiers are used to assess the validity of guide-plans, so why not simply use the plans to solve the MDPs outright? As mentioned in the discussion section:
> while we could use PDDL planners to get this abstract plan, these planners are useful in a narrow set of domains whereas LLMs can be useful in many more generalizable scenarios.

   However, I'm not convinced that this method was actually demonstrated to be more generalizable than PDDL planning. See point 4.
2. From what I can tell, all the environments are discrete tabular environments and there does not appear to be a mention of non-deterministic environments. Also, the environments do not appear particularly difficult or long-horizon.
3. The standard deviation appears to be huge in the RQ2 results, and the statistics are calculated over only 5 runs. This makes it difficult to compare the methods. I would suggest averaging over more runs.
4. The deterministic method relies on a huge amount of prompt engineering including a task description, an observation description, and a query description. This can include a textual representation of the entire environment, and in the hierarchical setting it includes a full PDDL specification. In addition, in the verifier condition the full transition dynamics are needed. The goal of reward shaping is to get an agent maximize reward in a difficult MDP that it doesn't have full access to. If you require nearly full access to the MDP to create the shaped reward function, what is the point of shaping? Perhaps a comparison to policy iteration or value iteration, which are given the entire MDP to calculate the optimal policy.
5. The deterministic <-> hierarchical dichotomy is ill-posed and doesn't fit nicely into the existing literature. There is a deterministic <-> non-deterministic dichotomy, and there can be determinism and non-determinism in a hierarchical setting. This choice of words is confusing. Long-horizon might be a better descriptor than hierarchical, and primitive or base-level might be a better descriptor of deterministic.

**Questions:**

1. Why are plan lengths and rewards averaged across 3 runs as mentioned in Table 1? What non-deterministic elements are you testing and why are 3 runs enough to get a good sample?
2. How were the hyperparameters chosen for the learning agents? Did they differ across experimental settings? There is no appendix attached.
3. Example environment prompts for the deterministic and hierarchical setting would be useful to the reader. Can we see an example?

---

> ### Author Response · Authors · 2024-11-21
> **Rebuttal**
>
> We thank the reviewer for their thoughtful comments and feedback, and that they find our idea novel and experiments to be comprehensive. We have tried addressing each of the concerns below:
>
> $$\textbf{Clarification on using LLM-generated plans (W1):}$$ Recall from lines 81-92 in the paper’s Introduction section where we talk about the underlying MDP to have stochastic transitions. In the deterministic abstraction case, we can not use the plan generated by the LLM to solve the task. Similarly, in the hierarchical abstraction case, the LLM-generated plan only consists of the right sequence of subgoals the agent has to achieve. Hence, in both cases, we construct the reward shaping function using the LLM-generated plans to accelerate the RL agent’s training for the downstream tasks.
>
> $$\textbf{Clarification on non-deterministic environments (W2):}$$ We do mention this in the Introduction section of the work (see response to W1), but have also reiterated this point for readers’ understanding again in the experiment section in lines 346 and 353..
>
> $$\textbf{High Standard Deviation in Figure 4 (W3):}$$ We double-checked our plots to find that there indeed was an error during the plot generation step in our results. We have updated our plots to fix the issue as can be seen in Figure 4 on page 9.
>
> $$\textbf{Efforts behind prompt engineering (W4):}$$ Since we aim to obtain the reward shaping plan from the LLM on the deterministic MDP, it becomes important to include information on the state and action space along with a description of the environment as part of the input prompt - a common practice in several existing works [3,4,5,6]. Moreover, we would like to highlight that this assumption is relaxed in the hierarchical MDP abstraction case where we prompt the LLM with the PDDL file which has been generated by the LLM itself [2], thereby reducing the human prompt effort significantly. An optimal policy can be helpful to construct the intended reward shaping function, as pointed out by the reviewer, but our work specifically looks at investigating the extent to which we can use off-the-shelf LLMs to provide any useful guidance for the task at hand. Note, that this can particularly be useful for environments which consist of an exponentially large state/action space where obtaining an optimal policy can be computationally expensive.
>
> $$\textbf{Clarification on paper’s taxonomy (W5):}$$ Our work does not intend to pose a dichotomy between a deterministic and a hierarchical MDP setup, but aims to study these two possible MDP abstractions for a downstream RL task which has sparse-reward and stochastic transitions. To further clarify this point and take the reviewer’s suggestion into consideration, we have updated the description of these two cases in our Introduction section in lines 93-95.
>
> We have tried addressing each of the questions below:
>
> $$\textbf{Clarification on Table 1 results (Q1):}$$ We do not intend to compare different LLMs but aim to show how effectively we can obtain a plan using the different LLMs with and without the presence of a verifier in Table 1 and 2. We average across 3 runs, however, note that we simply need one plan from the LLM for any given domain to construct the reward shaping function. Furthermore, we point the reviewer to the answer above to W2 to clarify the point on the non-determinism in the environments’ transitions.
>
> $$\textbf{Choice of hyperparameters (Q2):}$$ We ask the reviewer to check the attached zip file in our submission which contains the code and the appendix to our work. The hyperparameters have been chosen to be the same as our baseline in [1] and we have included these in Appendix C.2.
>
> $$\textbf{Example prompts and clarification on Appendix (Q3):}$$ We would like to point the reviewer to Section D of the appendix in the attached zip directory for all the prompts that we use in this work. We would be happy to answer any further questions regarding the examples.
>
> [1] Guan, Lin, Sarath Sreedharan, and Subbarao Kambhampati. "Leveraging approximate symbolic models for reinforcement learning via skill diversity." International Conference on Machine Learning. PMLR, 2022.
>
> [2] Lin Guan, Karthik Valmeekam, Sarath Sreedharan, and Subbarao Kambhampati. Leveraging pretrained large language models to construct and utilize world models for model-based task planning. Advances in Neural Information Processing Systems, 36:79081–79094, 2023.
>
> [3] Wang, Guanzhi, et al. "Voyager: An open-ended embodied agent with large language models." arXiv preprint arXiv:2305.16291 (2023).
>
> [4] Ma, Yecheng Jason, et al. "Eureka: Human-level reward design via coding large language models." arXiv preprint arXiv:2310.12931 (2023).
>
> [5] Yao, Shunyu, et al. "React: Synergizing reasoning and acting in language models." arXiv preprint arXiv:2210.03629 (2022).
>
> [6] Shinn, Noah, et al. "Reflexion: Language agents with verbal reinforcement learning." Advances in Neural Information Processing Systems 36 (2024).

---

> > ### Comment · Reviewer_JNWs · 2024-11-21
> >
> > I thank the authors for their response. While many of my questions were addressed, e.g. regarding determinism in the environment, the paper's taxonomy, hyperparameters used, I am unswayed by the authors' response to Weakness 4.
> >
> > After inspecting the textual descriptions of the environment used in the deterministic ("flat") MDP setting and looking at prior work, I believe that this is a reasonable assumption. However, in the hierarchical case, a full PDDL specification of the environment strikes me as perhaps too strong an assumption. As mentioned in your response to my critique, shaping can be useful in environments that consist of exponentially large state and action spaces for which computing an optimal policy is expensive. However, the environments in the hierarchical setting do not seem to be particularly difficult or complex, with relatively few actions and predicates. For these environments it is surely more efficient to use a classical planner to compute an optimal policy.
> >
> > Ultimately, if the authors claim that their method of reward shaping is more useful than planning in some class of domains, it is my belief that the authors should run experiments on such domains. In this specific case, it needs to be shown that LLMs can provide useful reward shaping advice for sufficiently complex environments, i.e. with sufficiently long PDDL specs. Furthermore, it is unclear weather reward shaping is necessary in such simple environments, as it seems that the LLM may be able to plan on its own to solve the environment using the given PDDL spec and the back-prompting verifier loop. I will increase the presentation score to "fair", but I will keep my overall score a 3 at present, though I welcome a response to these concerns.

---

> ### Author Response · Authors · 2024-11-24
> **Rebuttal**
>
> Thank you for the feedback! We would like to clarify the following points:
>
> 1) We do not assume access to the full PDDL specification for the hierarchical case for the following two reasons:
>
>      A) It is very common that a problem can not (or is challenging to) be fully represented with PDDL, for example problems such as in motion planning or problems that have stochastic dynamics (such as ours too).
>
>      B) The more complicated a PDDL is, the more likely that the model can be inaccurate or incomplete [1], and even humans can make mistakes as has also been noted in [2]. LLMs will naturally be more prone to errors.
>
> While the above two points are well known, here our standpoint is that reward shaping can circumvent many of the challenges posed above. For our hierarchical case, the PDDL specification only consists of sub-goals that the downstream agent needs to achieve. Our results for RQ2 also indicate that the RL agent is able to perform better than the baseline Q-learning even when a partial set of subgoals is present in the PDDL specification.
>
> Moreover, the underlying MDP is stochastic even in the hierarchical case. For a classical planner to compute an optimal policy for the underlying MDP, a human (domain expert) or some other information source needs to provide a complete PDDL specification taking stochastic transitions into account, which is not trivial.
>
> 2) The idea behind using concise PDDL specifications with respect to the underlying MDP is that we are able to generate a reward shaping function using this LLM-generated specification. Naturally, having a more complex PDDL specification that consists of the entire low-level action space and state constraints (as preconditions and effects) will lead to a stronger reward shaping function that can potentially guide the RL agent at every step. However, it is not feasible to obtain such complex PDDL specifications other than asking from domain experts as has been stated above.
>
> We hope our clarification is helpful in understanding the hierarchical abstraction setting. We would be happy to answer any further questions!
>
>
> [1] Guan, Lin, Sarath Sreedharan, and Subbarao Kambhampati. "Leveraging approximate symbolic models for reinforcement learning via skill diversity." International Conference on Machine Learning. PMLR, 2022.
>
> [2] Lin Guan, Karthik Valmeekam, Sarath Sreedharan, and Subbarao Kambhampati. Leveraging pretrained large language models to construct and utilize world models for model-based task planning. Advances in Neural Information Processing Systems, 36:79081–79094, 2023.

---

> > ### Comment · Reviewer_JNWs · 2024-11-26
> >
> > Thank you for clarifying that you do not always use a full PDDL specification.
> > - Regarding 1A, I will note that there is PPDDL -- which can handle probabilistic planning environments -- and if you want to argue for shaping in a motion planning domain, you should use your method in at least one motion planning domain.
> > - Regarding 1B, this is actually a summary of one of my objections. If your goal is to show that reward shaping using an LLM is more useful than planning (even probabilistically) in large domains, you must actually test your method in such a domain. If you believe that LLMs may struggle with more complicated PDDL specs, that is a knock on your method, which you have suggested will be helpful for more complex domains.
> >
> > Can you please elaborate on this?
> > >For our hierarchical case, the PDDL specification only consists of sub-goals that the downstream agent needs to achieve.
> >
> > I don't think I saw these in the appendix, it seemed as though there was only a single predicate for each PDDL goal. What kind of subgoals are you specifying?

---

### Official Review · Reviewer_komg · 2024-11-03

**Soundness:** 2
**Presentation:** 2
**Contribution:** 2
**Rating:** 3
**Confidence:** 4

**Summary:**

This paper focuses on the challenge of sample inefficiency in reinforcement learning (RL) in sparse reward domains, particularly in the presence of stochastic transitions. The authors propose using Large Language Models (LLMs) to generate heuristics for constructing a reward shaping function that can enhance the sample efficiency of RL agents. Specifically, they leverage off-the-shelf LLMs to generate a plan for an abstraction of the underlying MDP. They conduct experiments in various domains, demonstrating the improved performance of RL algorithms when guided by LLM-generated heuristics.

**Strengths:**

- The idea of leveraging heuristics fom LLMs for enhancing the sample efficiency of reinforcement learning is interesting and promising.

**Weaknesses:**

- My main concern is the contribution novelty. This work heavily relies on the PDDL model introduced in prior research [1]. The use of LLM verifiers or feedback from environments to revise plans is not a novel idea [2, 3].
- The paper lacks a detailed description of the extraction of the PDDL models and the reward shaping functions, which induces a challenging reading experience.
- The experimental domains are limited to grid-world settings with discrete state and action spaces that have explicit semantics, raising significant concerns about scalability and applicability.
- The introduction of the evaluation benchmarks in Appendix B lacks details about the observation space, making it difficult to assess whether the proposed method is applicable to tasks with partial observability.
- There are no citations regarding the evaluation benchmarks, making it hard to ascertain the representativeness of the evaluation domains, especially for Mario and Minecraft, as their descriptions in the paper do not align with common understandings.
- There is a lack of comparisons with other LLM-enhanced RL methods. For example, the authors mention several related works in Section 2, but none are considered for comparison in the experiments.
- What is the inference frequency of the LLMs in providing the guide plan? It seems that the proposed method incurs significantly higher LLM inference costs, raising concerns about its practical efficiency.
- It appears that the proposed method requires substantial manual effort to design the prompts, further raising concerns about its real efficiency compared to manually designed reward functions or human-provided plans.
- There is limited discussion of the paper's limitations.
- Why is only a tabular Q-learning method utilized in RQ2.2? What is the performance of deep RL methods, such as DQN or PPO?

[1] Lin Guan, Karthik Valmeekam, Sarath Sreedharan, and Subbarao Kambhampati. Leveraging pretrained large language models to construct and utilize world models for model-based task planning. Advances in Neural Information Processing Systems, 36:79081–79094, 2023.

[2] Shinn N, Cassano F, Gopinath A, et al. Reflexion: Language agents with verbal reinforcement learning[J]. Advances in Neural Information Processing Systems, 2024, 36.

[3] Zhu X, Chen Y, Tian H, et al. Ghost in the minecraft: Generally capable agents for open-world environments via large language models with text-based knowledge and memory[J]. arXiv preprint arXiv:2305.17144, 2023.

**Questions:**

Please take a look at the Weaknesses section.

---

> ### Author Response · Authors · 2024-11-21
> **Rebuttal**
>
> We thank the reviewer for their thoughtful comments and feedback, and that they find our idea promising. We have tried addressing each of the concerns below:
>
> $$\textbf{Contribution novelty (W1):}$$ To reiterate on the claims of our work, we would like to highlight from lines 102-112 in the paper that we aim to study the effectiveness of using LLMs for solving simple problem abstractions of downstream sparse-reward RL problems with stochastic transitions. While we obtain a PDDL model of the underlying MDP problem as in [1] and show the effectiveness of the guidance obtained using LLMs with and without a verifier, we go further and show a novel approach for utilizing these LLM-generated plans on abstract MDPs to construct reward shaping function for boosting the sample efficiency of downstream RL agents.
>
> $$\textbf{Extraction of PDDL models (W2):}$$ We regret the confusion regarding the PDDL extraction step as we replicate the steps used in [1]. Furthermore, we would like to point the reviewer to the example prompts in Appendix section D for reference. We would be happy to answer any further clarifications.
>
> $$\textbf{Scalability (W3 and W4):}$$ We restrict the current scope of the work to discrete action space environments with stochastic transitions and present a comprehensive analysis on multiple environments using two different MDP abstractions. However, we believe that extending our reward shaping work to partially observable action space environments can be a valuable future extension. For example, our hierarchical abstraction, which requires LLMs to output the right sequence of subgoals that the agent has to achieve, can be easily utilized. Similar to the discrete action space experiments shown for Mario, Minecraft, and the Household environment in our work, the RL agent can learn a policy that follows the correct sequence of subgoals using shaped rewards in a partially observable environment too. We have also included this point in our Discussion section in lines 510-515.
>
> $$\textbf{Clarification on Mario and Minecraft (W5):}$$ We do point the reviewer to lines 409-413 where we have cited the relevant work from where we adopt the environments and the baselines.
>
> $$\textbf{Comparison to other LLM-enhanced RL methods (W6):}$$ Due to the fundamental differences in our approach, we do not consider baseline from the works that we mention in the Related Works section. None of the existing works utilize LLMs to obtain guidance for an underlying sparse-reward stochastic environment on which we intend to boost the RL agent’s sample efficiency.
>
> $$\textbf{Clarification on LLM inference costs (W7):}$$ The total cost to complete all our LLM experiments (with multiple runs) for Table 1 and Table 2 results combined was below $20.
>
> $$\textbf{Effort behind prompt designs (W8):}$$ Since we aim to obtain the reward shaping plan from the LLM on the deterministic MDP, it becomes important to include information on the state and action space along with a description of the environment as part of the input prompt - a common practice in several popular works [3, 4, 5, 6]. Moreover, we would like to highlight that this assumption is relaxed in the hierarchical MDP abstraction case where we prompt the LLM with the PDDL file which has been generated by the LLM itself [1], thereby reducing the human prompt effort significantly.
>
> $$\textbf{Limitations of the proposed approach (W9):}$$ We have highlighted some of the important limitations and trade-offs to be considered for our approach in the Discussion section 5.4. We would be happy to answer any further clarifications.
>
> $$\textbf{Q-learning algorithm for RQ 2.2 (W10):}$$ For the sake of uniform comparisons in our empirical setting, we use the algorithm exactly as in our baseline in [2].
>
> [1] Lin Guan, Karthik Valmeekam, Sarath Sreedharan, and Subbarao Kambhampati. Leveraging pretrained large language models to construct and utilize world models for model-based task planning. Advances in Neural Information Processing Systems, 36:79081–79094, 2023.
>
> [2] Guan, Lin, Sarath Sreedharan, and Subbarao Kambhampati. "Leveraging approximate symbolic models for reinforcement learning via skill diversity." International Conference on Machine Learning. PMLR, 2022.
>
> [3] Wang, Guanzhi, et al. "Voyager: An open-ended embodied agent with large language models." arXiv preprint arXiv:2305.16291 (2023).
>
> [4] Ma, Yecheng Jason, et al. "Eureka: Human-level reward design via coding large language models." arXiv preprint arXiv:2310.12931 (2023).
>
> [5] Yao, Shunyu, et al. "React: Synergizing reasoning and acting in language models." arXiv preprint arXiv:2210.03629 (2022).
>
> [6] Shinn, Noah, et al. "Reflexion: Language agents with verbal reinforcement learning." Advances in Neural Information Processing Systems 36 (2024).

---

### Official Review · Reviewer_Bu43 · 2024-11-04

**Soundness:** 3
**Presentation:** 3
**Contribution:** 2
**Rating:** 5
**Confidence:** 4

**Summary:**

This paper aims to answer the following question: Can we obtain heuristics using LLMs for constructing a reward shaping function that boost an RL agent's sample efficieny? To this end, authors propose to use off-the-shelf LLMs to generate a plan for an abstraction of the underlying MDP, which will then be used as a heuristic to construct the reward shaping signal for the downstram RL algrithms (e.g., PPO, A2C, etc.). Experiments on multiple tasks, including Baby AI environment suite, Household, Mario and Minecraft domain illustrates the effectiveness of the poposed method.

**Strengths:**

1. This paper is clearly written and easy to follow.

**Weaknesses:**

1. [1] focuses on the same problem, i.e., using an LLM for reward shaping to improve the sample efficiency of RL algorithms. It also proposes to use goal-based potential shaping. However, relevant comparisions and discussions are missing.
2. The verifier in the proposed framework should be manully designed, which is task-specific and rule-based, or even burdensome for humans to design.

[1] Zhang, Fuxiang, et al. "Improving Sample Efficiency of Reinforcement Learning with Background Knowledge from Large Language Models." arXiv preprint arXiv:2407.03964 (2024).

**Questions:**

1. In all of the environments, the action space are discrete and has only one dimension. What about multi-dimensional continuous action space or even hybrid action space (e.g., turn right 34 degrees)?
2. For the long-horizon stochastic environments, the LLM will generate a high-level plan, in which a high-level state will serve as a sub-goal
for reward shaping. How can we determine which sub-goal to use? How can we ensure that a sub-goal is finished?

---

> ### Author Response · Authors · 2024-11-21
> **Rebuttal**
>
> We thank the reviewer for their thoughtful comments and feedback, and that they find our paper easy to follow. We have tried addressing each of the concerns below:
>
> $$\textbf{Comparison to referenced paper (W1):}$$ A very important difference between the referenced work by Zhang et al. and our work is in the problem setting since we are trying to focus on sparse-reward problems with stochastic transitions. Due to a stochastic transition function in the underlying MDP, the problem of sample inefficiency is pronounced. Hence, our solution involves constructing an abstraction of this underlying MDP where we study the use of LLMs to extract shaped rewards. Nevertheless, since this work also falls under the direction of using LLMs for RL problems, we apologize for having missed this and have now included it in our Related Works section in line 135 where we talk about other RL works that focus on LLM-based guidance.
>
> $$\textbf{Assumptions behind constructing the verifier (W2):}$$ We would like to highlight that in the hierarchical abstraction case, the PDDL (symbolic) model of the task is teased out of the LLM, this assumption behind the verifier is relaxed (as mentioned in lines 511-520 in the paper). Note, that this is dependent on how expensive it may be to obtain a domain model for the task at hand, and if there are any constraints on the RL agent's training stage. Our results consistently show that having a verifier in the loop can significantly improve the quality of the obtained heuristics which can boost RL agent’s sample efficiency.
>
>
> We have tried addressing each of the questions below:
>
> $$\textbf{Extending approach to complex action-space environments (Q1):}$$ We restrict the current scope of the work to discrete action space environments with stochastic transitions and present a comprehensive analysis on multiple environments using two different MDP abstractions. However, we believe that extending our reward shaping work to continuous action space environments can be a valuable future extension. For example, our hierarchical abstraction, which requires LLMs to output the right sequence of subgoals that the agent has to achieve, can be easily utilized for continuous action spaces. Similar to the discrete action space experiments shown for Mario, Minecraft, and the Household environment in our work, the RL agent can learn a policy that follows the correct sequence of subgoals using shaped rewards in a continuous environment too. We have also included this point in our Discussion sub-section in lines 510-515.
>
> $$\textbf{Subgoal verification (Q2):}$$ Currently, we generate the PDDL files by prompting the LLM, as has been proposed in [1], where the work relies on the assumption that LLMs can rely on their background training data for generating the subgoals for respective domains. To ensure that the subgoal has been reached, we can either extract that information from the environment state as has been done in our baseline [2] or train binary classifiers as has been done in [3], or even use state-of-the-art VLMs [4].
>
> [1] Lin Guan, Karthik Valmeekam, Sarath Sreedharan, and Subbarao Kambhampati. Leveraging pretrained large language models to construct and utilize world models for model-based task planning. Advances in Neural Information Processing Systems, 36:79081–79094, 2023.
>
> [2] Guan, Lin, Sarath Sreedharan, and Subbarao Kambhampati. "Leveraging approximate symbolic models for reinforcement learning via skill diversity." International Conference on Machine Learning. PMLR, 2022.
>
> [3] Mirchandani, Suvir, Siddharth Karamcheti, and Dorsa Sadigh. "Ella: Exploration through learned language abstraction." Advances in neural information processing systems 34 (2021): 29529-29540.
>
> [4] Du, Yuqing, et al. "Vision-language models as success detectors." arXiv preprint arXiv:2303.07280 (2023).

---

### Official Review · Reviewer_uAhh · 2024-11-10

**Soundness:** 3
**Presentation:** 2
**Contribution:** 2
**Rating:** 3
**Confidence:** 4

**Summary:**

The paper addresses the challenge of sample inefficiency in RL, especially in sparse reward settings and domains with stochastic transitions. It proposes using heuristics derived from LLMs to shape rewards and improve RL training efficiency. The authors explore the idea of leveraging LLMs to generate plans that act as heuristics for constructing reward-shaping functions, which can help RL agents converge more quickly to optimal policies. Experiments across diverse environments (BabyAI, Household, Mario, Minecraft) demonstrate improvements in sample efficiency.

**Strengths:**

1. The overall algorithm is straightforward and easy to understand.
2. Experiments are conducted under different settings and environments and the discussion of the experimental results is clear.

**Weaknesses:**

1.	The LLM prompting part is too simple. According to the illustration, all prompting parts are just necessary to input all state and task information for LLM to process. Also, the verifier part is more like a legal action checker, which is also common in a lot of LLM agent implementations and frameworks. Considering this, the writing illustration is over-complicated and the idea of the prompting part does not preserve too much novelty.
2.	The writing is not that clear and makes me confused. For example, the figure 1 seems to be confusing. It mentions that the algorithm updates the buffer with shaped rewards, but there is no arrow pointed from the replay buffer. Only an arrow points to the replay buffer. (see more in questions)
3.	The experiment results cannot fully support the argument. According to the result in Table 1, it seems that GPT-3.5 and GPT-4o themselves can already reach a high reward (e.g., 0.9+ for empty room, 0.85+ for LavaGap). Then what is the sense of training RL with these expert-level LLMs? Why not directly imitate these LLM’s policies? Also, the paper needs to take more ablation studies (how different setting influences the performance) and comparisons with other LLM+RL algorithms (for example, [1] also conducted experiments on BabyAI)

[1] Srivastava, Megha, Cedric Colas, Dorsa Sadigh, and Jacob Andreas. "Policy Learning with a Language Bottleneck." arXiv preprint arXiv:2405.04118 (2024).

**Questions:**

1.	What specifically is the potential function used in the reward shaping?
2.	Which LLM are you using in Section 5.3?
3.	Why does figure 4 have such a big standard deviation in all 3 tasks?
4.	What is the major limitation of your algorithm?

---

> ### Author Response · Authors · 2024-11-21
> **Rebuttal**
>
> We thank the reviewer for their thoughtful comments and feedback, and that they find our algorithm easy to understand and the experiment section comprehensive. We have tried addressing each of the concerns below:
>
> $$\textbf{Prompting and paper’s claims (W1):}$$ Our primary contributions in this work do not include claiming any novelty regarding the prompting step or the check made using the verifier for legal moves. As stated in lines 102-112 of the paper, we focus on studying how well LLMs can solve simple problem abstractions of sparse-reward RL tasks with stochastic transitions. We are actually glad that the prompting step is simple enough, as verified by the reviewer.
>
> $$\textbf{Clarifications on writing and figure (W2):}$$ Once we have the desired <state, action> pairs (in the deterministic abstraction setting) or the desired <state>s (in the hierarchical abstraction setting) from prompting the LLM, we simply store this information in Step II (as shown in Figure 1). During the RL training process, we check for these desired <state, action> pairs or <state>s in the respective settings and reward shape those transitions. Hence, we did not show an arrow from the replay buffer in Figure 1.
>
> $$\textbf{Clarification on experiment results (W3):}$$ Recall from lines 81-92 in the paper’s Introduction section, that we talk about the underlying MDP to have stochastic transitions. In the deterministic abstraction case, we can not use the plan generated by the LLM to solve the task. Similarly, in the hierarchical abstraction case, the LLM-generated plan only consists of the right sequence of subgoals the agent has to achieve. Hence, in both cases, we construct the reward shaping function using the LLM-generated plans to accelerate the RL agent’s training for the downstream tasks.
>
> About the paper the reviewer referenced, it talks about using LLMs to generate linguistic rules that can be helpful to solve two-player games. The experiments in the paper include two-player communication, maze, and image-generation tasks. However, we do not see any experiments on the BabyAI domain in the current arxiv version of the work (https://arxiv.org/pdf/2405.04118). Moreover, the problem setup is very different from ours where the referenced work aims to generate interpretable (and thus, linguistic) rules that allow for solving the two-player communication tasks. We focus on the single-agent setting where the objective is to accelerate RL agent’s training on a sparse-reward problem with stochastic transitions. We would be happy to answer any further questions the reviewer may have regarding the setup or comparisons between the two works, if we missed anything. Moreover, since this work also falls under the overall direction of using LLMs for decision-making problems, we have included this work in our Related Work section in line 135 where we talk about other RL works that focus on LLM-based guidance.

---

> > ### Comment · Reviewer_uAhh · 2024-11-28
> >
> > Thanks for the response.
> >
> > * Regarding w1, can you elaborate more on your contributions? If the prompting part is not one of them, what is the main contribution of the technical part? Moreover, it also weakens your technical contribution to spending more than 1 page (and consider the whole sec 4 are just two pages) on something that is not your contribution?
> > * Regarding w3, I still didn't get the point. Why we cannot use the LLM's result In the deterministic abstraction case? If not what the avg reward stands for in the table 1. I understand sometimes they can only generate partial solution, but I assume if you obtain a reward signal, that means that it generate the full paths?
> >
> > And my concerns in the question section have not been addressed.
> >
> > I maintain my score.

---

### Note · Authors · 2024-12-05

I have read and agree with the venue's withdrawal policy on behalf of myself and my co-authors.